# Is Pathologic Axillary Staging Valid If Lymph Nodes Are Less than 10 with Axillary Lymph Node Dissection after Neoadjuvant Chemotherapy?

**DOI:** 10.3390/jcm11216564

**Published:** 2022-11-05

**Authors:** Hee Jun Choi, Jai Min Ryu, Jun Ho Lee, Yoonju Bang, Jongwook Oh, Byung-Joo Chae, Seok Jin Nam, Seok Won Kim, Jeong Eon Lee, Se Kyung Lee, Jonghan Yu

**Affiliations:** 1Department of Surgery, Samsung Changwon Hospital, Sungkyunkwan University School of Medicine, Changwon 51353, Korea; 2Division of Breast Surgery, Department of Surgery, Samsung Medical Center, Sungkyunkwan University School of Medicine, Seoul 06351, Korea

**Keywords:** neoadjuvant chemotherapy, number of lymph nodes, axillary lymph node dissection

## Abstract

Introduction: The aim of this study was to evaluate the prognostic value of the number of lymph nodes removed in breast cancer patients who undergo axillary lymph node dissection (ALND) after neoadjuvant chemotherapy (NAC). Methods: We included patients who were diagnosed with invasive breast cancer and cytology with proven involved axillary node metastasis at diagnosis and treated with NAC followed by curative surgery at Samsung Medical Center between January 2007 and December 2015. The primary outcomes were disease-free survival (DFS) and overall survival (OS). Results: Among 772 patients with NAC and ALND, there were 285 ypN0, 258 ypN1, 135 ypN2, and 94 ypN3 cases. The median follow-up duration was 69.0 months. The group with less than 10 lymph nodes number (<10 nodes group) included 123 patients and the group with 10 or more lymph nodes number (≥10 nodes group) included 649 patients. There were no significant differences in DFS (*p* = 0.501) or OS (*p* = 0.883) between the two groups. In the ypN0 subgroup, the <10 nodes group had worse DFS than ≥10 nodes group (*p* = 0.024). In the ypN1 subgroup, there were no significant differences in DFS (*p* = 0.846) or OS (*p* = 0.774) between the two groups. In the ypN2 subgroup, the <10 nodes group had worse DFS (*p* = 0.025) and OS (*p* = 0.031) than ≥10 nodes group Conclusion: In ypN0 and ypN2 subgroups, breast cancer patients with less than 10 lymph nodes number in ALND after NAC might be considered for additional staging or closer surveillance when compared to patients with 10 or more than lymph node.

## 1. Introduction

Neoadjuvant chemotherapy (NAC) is an important component of combination therapy strategies for advanced breast cancer. The effectiveness to chemotherapy can be predicted and the breast conservation rate in patients with breast cancer can be improved [1,2]. Axillary lymph node metastasis is still one of the most important factors in breast cancer and is useful for guiding treatment.

Axillary lymph node dissection (ALND) is an important method for evaluating axillary node status in breast cancer. The extent of ALND and the number of nodes needing to be removed has been controversial. Generally, breast cancer patient survival is improved by removing more axillary lymph nodes. Previous studies have suggested that in node-positive breast cancer, the number of nodes retrieved is significantly associated with an increased positive nodal count and greater use of adjuvant therapy [3].

Recent studies have shown that ALND may be omitted if the axillary response after NAC is good. Park et al. showed the false negative rate of sentinel lymph node biopsy (SLNB) after NAC was 7.8% and there was no difference in overall survival (OS, *p* = 0.2) or the regional recurrence-free survival (*p* = 0.297) between SLNB and ALND groups [4]. In the SENTINA (sentinel lymph node biopsy in patients with breast cancer before and after NAC) trial, ALND could be replaced with SLNB when at least three lymph nodes are identified [5]. As the NAC effect gradually improves, the application of ALND decreases [6]. However, ALND remains an important surgical method for oncological safety. Another study reported that the number of removed lymph nodes after NAC decreased and 16.3% of patients with NAC had less than 10 lymph nodes [7]. Thus, the decrease in the number of removed lymph nodes can underestimate the nodal stage.

For the nodal stage, we typically require more than 10 lymph nodes [8]; however, we have experienced less than 10 nodes excised in ALND after NAC. The aim of this study was to evaluate the prognostic value of the number of lymph nodes in breast cancer patients who undergo ALND after NAC.

## 2. Materials and Methods

This study was a medical record review based on a prospectively collected cohort. We included 772 patients who were diagnosed with invasive breast cancer, cytology proven axillary node positive, and treated with NAC followed by curative surgery at Samsung Medical Center between January 2007 and December 2015 (Figure 1). Inclusion criteria were patients who (1) were female with unilateral breast cancer, (2) were cytology proven axillary node positive before NAC, (3) completed all intended cycles of NAC before surgery, (4) underwent ALND after NAC, (5) had complete surgical resection of the tumor and negative surgical margins. We excluded any cases of metastatic breast cancer, bilateral breast cancer and patients who did not properly perform endocrine treatment or target therapy. Two groups were analyzed for the study, one group had less than 10 lymph nodes (<10 nodes group) after ALND and the other had 10 or more lymph nodes (≥10 nodes group). The DFS was defined as the time interval between date of diagnosis and date of first recurrence or last follow-up or death, whichever came first. The OS was defined as the time interval between date of diagnosis and date of last follow-up or death.

Most patients received anthracycline- and/or taxane-based regimens. These regimens included anthracycline plus cyclophosphamide, followed by anthracycline-based, taxane-based, or trastuzumab regimens. The adjuvant radiotherapy (RT) was performed in all patients with breast conserving surgery or lymph node metastasis. The indications of RT to supraclavicular fossa were T3 or higher, N2 or higher, or supraclavicular node metastasis confirmed by FNA. The indications of RT to internal mammary chain are metastasis suspected by breast MRI or internal mammary node metastasis confirmed by FNA.

We used the chi-square test and Spearman’s correlation coefficient to compare discrete variables, and we conducted survival analysis with the log-rank test. The Kaplan–Meier method was used to draw survival curves to statistically determine the significance related to survival. Differences were assumed to be significant when the *p*-value was less than 0.05. We used SPSS, version 22 (IBM Corp., Armonk, NY, USA) for the chi-square tests and for calculating logistic regression. The Institutional Review Board of Samsung Medical Center, Seoul, Korea (IRB file no.2017-09-051) approved this study.

## 3. Results

Among 772 patients treated with NAC and ALND, the <10 nodes group included 123 patients and the ≥10 nodes group included 649 patients. There were 285 ypN0, 258 ypN1, 135 ypN2, and 94 ypN3 cases. The median follow-up was 69.0 months. Table 1 shows the characteristics of patients according to lymph node number. The <10 nodes group had alower BMI, radiotherapy recipient rate, and pathologic stage than the ≥10 nodes group. The <10 nodes group had 29.2% breast pCR and 56.1% axillary pCR. The ≥10 nodes group had 19.1% breast pCR and 33.3% axillary pCR. The proportion of patients with residual nodal burden was 43.9% in the <10 node group, and 66.7% in the ≥10 nodes group. The proportion of positive nodes out of total nodal harvest was 1.4/8.6 positive nodes (16.3%) in the <10 nodes group and 2.2/14.7 in the ≥10 nodes group (15.0%). In the <10 nodes group, there were 31 recurrences, of which 16 (breast recurrence: 4, axilla recurrence: 12) had local recurrence. In the ≥10 nodes group, 153 had recurrence, of which 68 (breast recurrence: 14, axilla recurrence: 53, both recurrence: 1) had local recurrence. There was no significant difference in disease-free survival (DFS, *p* = 0.949) and overall survival (OS, *p* = 0.734) between the <10 nodes group and the ≥10 nodes group. After compensating ypT and ypN staging, there was a strong trend in DFS; however, there was no significant difference in DFS (*p* = 0.056) and OS (*p* = 0.528) between the <10 nodes group and the ≥10 nodes group (Figure 2). There were no significant difference in DFS(*p* = 0.972) and OS (*p* = 0.758) in LN positive patients (Figure 3).

Among the 285 ypN0 patients, the <10 nodes group included 69 patients and the ≥10 nodes group included 216 patients. There were 13 recurrences in the <10 lymph nodes group, of which 5 had local recurrence, 26 had recurrence in the ≥10 lymph nodes group, and 7 had local recurrence. Table 2 shows the characteristics of the pathologic node-negative ALND group according to removed lymph node number. The ypN0 group with <10 nodes had no significantly different characteristics compared with the ypN0 group with ≥10 nodes. In the ypN0 subgroup, the <10 nodes group had worse DFS than the ≥10 nodes group (*p* = 0.024). There was no significant difference in OS between the <10 nodes group and the ≥10 nodes group (*p* = 0.818) (Figure 4A).

Table 3 shows the characteristics of the pathologic node-positive ALND group according to removed lymph node number. The ypN positive group with <10 nodes had no significantly different characteristics compared with the ypN positive group with ≥10 nodes.

Among the 258 ypN1 subgroup, the <10 nodes group included 39 patients and the ≥10 nodes group included 219 patients. There was no significant difference in DFS (*p* = 0.846) and OS (*p* = 0.774) between the <10 nodes group and the ≥10 nodes group in the ypN1 subgroup (Figure 4B).

Among the 135 ypN2 subgroup, the <10 nodes group included 15 patients and the ≥10 nodes group included 120 patients. The <10 nodes group had worse DFS (*p* = 0.025) and OS (*p* = 0.031) than the ≥10 nodes group in the ypN2 subgroup (Figure 4C).

## 4. Discussion

We applied lymph node number analyses to a cohort of patients who received NAC before curative surgery with cytology proven axillary metastasis. The adequacy of axillary surgery is important for proper staging of the axilla [9]. For evaluating node status, we typically need more than 10 lymph nodes. In this study, there was no significant difference in DFS or OS between the <10 nodes group and the ≥10 nodes group for all patients. However, in the ypN0 and ypN2 subgroup, the <10 nodes group had worse DFS than the ≥10 nodes group. There is a notable increase in ypN2 and ypN3 in the ≥10 nodes patients. In the group with the <10 lymph nodes, there is no ypN3 and it is possible that axillary staging was underestimated overall. The hypotheses to explain the inferior outcomes observed in this study where that fewer axillary LNs were obtained, including(1) that a lower LN population allows a false nodal stage,(2) that involved LNs may have been left behind by less complete surgery, and (3) that missing positive nodes leads to less post-op adjuvant therapy.

According to the breast cancer subtype, triple negative breast cancer (TNBC) and HER2-positive breast cancer are much more likely to achieve a pathological complete response (pCR) to NAC than HR-positive cancers [10,11,12]. In our study, there was no breast cancer subtype difference in the number of removed lymph nodes. In addition, this study contained only patients with cytology proven axillary node-positive breast cancer before NAC and show axillary pCR to NAC was higher than breast pCR.

Lymph node status is a very important prognosis factor in breast cancer with NAC [13,14]. Residual disease after NAC in lymph nodes at the time of definite surgery portends poor prognosis. The number of dissected lymph nodes from patients with breast cancer having NAC may be less than in patients without NAC, which may not always be carried out with an adequate ALND. Uyan et al. showed that median number of ALND with NAC was 16, while it was 20 without NAC and 16.3% of patients with NAC had<10 lymph nodes [7]. Insufficient lymph node dissection may result in inaccurate lymph node staging, and removing more lymph nodes makes for a more accurate determination of lymph node status [15]. Therefore, ALND with the <10 lymph nodes after NAC may be considered to be incomplete due to its poor accuracy. In our study, in some subgroups, the <10 nodes group had a poor oncological outcome compared with the ≥10 nodes group

Several studies evaluated the prognostic value of the number of negative lymph nodes (NLNs) in breast cancer patients after mastectomy [16]. Wu et al. showed that the DFS of patients with >10 NLNs was significantly higher than that of patents with ≤10 NLNs, and the 5-year DFS rates were 87.5% and 69.5%, respectively (*p* < 0.001) [17]. Karlsson et al. reported that the number of NLNs removed was an independent factor affecting prognosis, where patients with ≥10 NLNs removed had a better prognosis than patients with <10 NLNs removed, which affected node-positive patients but not node negative patients [18]. In our study, we surveyed the number of total lymph nodes removed instead of the number of NLMs. The <10 nodes group had worse DFS than the ≥10 nodes group in the ypN0 subgroup. The <10 nodes group might harbor occult, unresected positive lymph nodes.

One study focusing on the lymph node ratio (LNR) in breast cancer demonstrated that significant associations between LNR and DFS were found in hormone receptor-positive (*p* = 0.02) and TNBC (*p* = 0.003) subtypes [19]. The LNR is described as the ratio of number of positive ALN to total number of ALND. Another study demonstrated that LNR was an independent prognostic factor for DFS and OS (*p* < 0.05), while the ypN stage had no effect on prognosis (*p* > 0.05) [20].

A current clinical trial on Alliance A011202, which will provide SLNB information after NAC on clinical T1-3N1M0 breast cancer, was presented at the 2017 San Antonio breast cancer symposium. In this trial, the patients who have a pCR in the lymph nodes can have ALND omitted and patients who do not achieve a pCR in the lymph nodes can be compared to assess whether ALND will improve recurrence rates compared with no further axillary surgery and radiotherapy. Our study shows that among the ypN2 subgroup, the <10 nodes group had worse DFS (*p* = 0.025) and OS (*p* = 0.031) than the ≥10 nodes group. The <10 nodes group may have a higher axillary stage reality than is apparent from the more limited surgery. Another study compared groups where 1–7, 8–22, and >22 LNs were harvested after NAC. The better survival was independently associated with retrieval of up to approximately 20 LN [3]. In our study, 10 lymph nodes were divided based on the criteria of N3 [8], and in the case of ypN2, there was also a difference in the survival rate. This study showed that worse survival was associated with retrieving fewer LNs, likely as a result of an under-staged axilla and missed opportunity for adjuvant therapy. The ≥10 LN group had higher LN burdens. We presume positive nodes are easier to find at surgery but it then should have the opposite effect on outcome. There is more chance after NAC, that distribution of LN involvement may be more variable due to variable response in different nodes.

This study has a few limitations. First, the study was performed in a single comprehensive cancer institution in Korea. Second, the number of patients was relatively small. Third, oncology survival is dependent on a number of factors such as type of systemic therapy, compliance with adjuvant treatment, type of adjuvant treatment and rate of pCR/residual cancer burden after NAC. This study was not a prospective randomized clinical trial, so the distribution of patients might be uneven and had some effect on the results of regional control. Nevertheless, this study had a median follow-up time of 69 months, and the results provide important insight into the significance of the total lymph node number obtained at ALND following NAC.

In conclusion, breast cancer patients with less than 10 lymph nodes in ALND after NAC might be considered for additional staging or closer surveillance when compared to patients with 10 or more lymph nodes in ypN0 and ypN2 subgroups. We will need further validation from prospective trials with a broader spectrum of variables which potentially impact oncology survival.

## Figures and Tables

**Figure 1 jcm-11-06564-f001:**
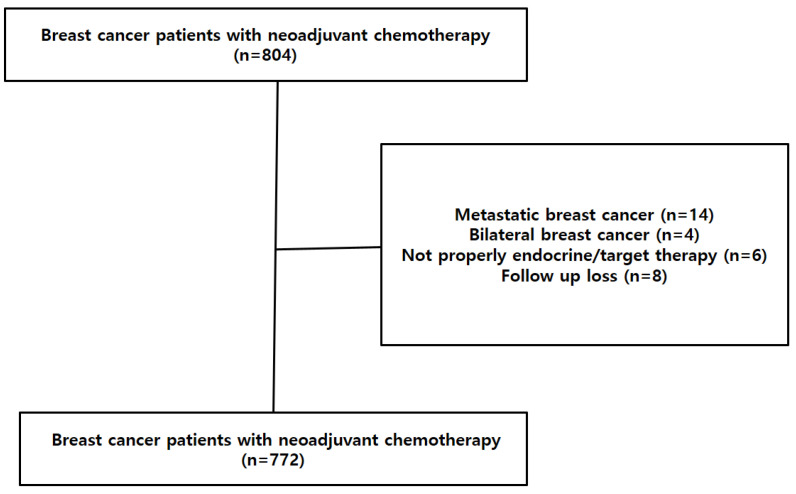
Consort diagram of patient selection with neoadjuvant chemotherapy.

**Figure 2 jcm-11-06564-f002:**
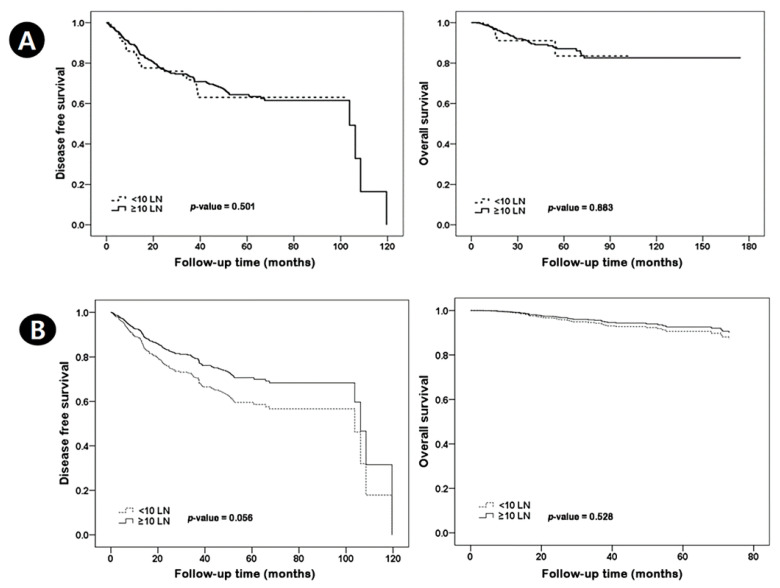
Kaplan–Meier survival curves for disease-free survival and overall survival according to lymph node number. Overall (**A**), compensating for ypT and ypN staging (**B**).

**Figure 3 jcm-11-06564-f003:**
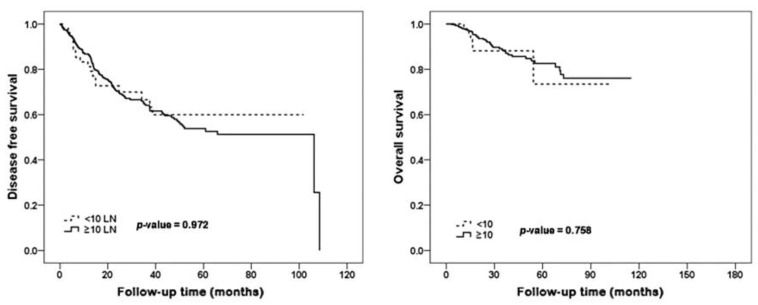
Kaplan–Meier survival curves for disease-free survival and overall survival in lymph node-positive patients according to lymph node number.

**Figure 4 jcm-11-06564-f004:**
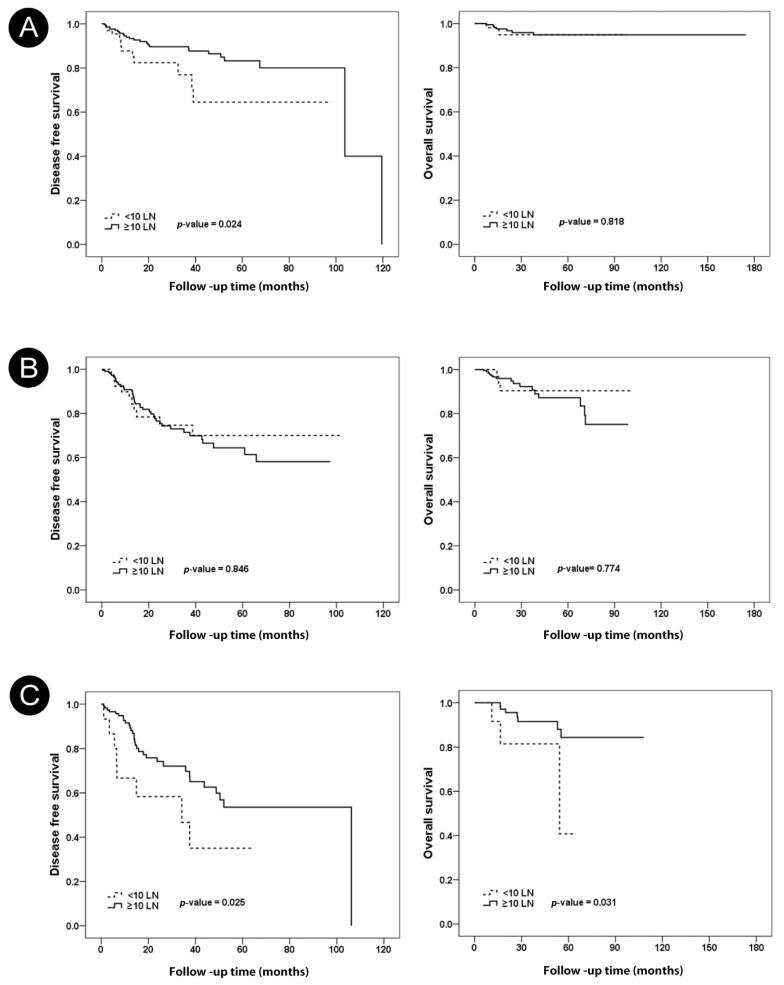
Impact of the number of lymph nodes on disease-free survival and overall survival of ypN0 (**A**), ypN1 (**B**), ypN2 patients (**C**).

**Table 1 jcm-11-06564-t001:** Characteristics of the patients’ group according to lymph node number.

Variable	<10 Nodes (*n* = 123) No. (%)	≥10 Nodes (*n* = 649) No. (%)	Total (*n* = 772) No. (%)	*p*-Value
Age (years)	45.8 ± 10.1	45.1 ± 9.4	45.2 ± 9.5	0.419
BMI	23.2 ± 3.1	24.1 ± 3.4	23.9 ± 3.4	0.011
Type of surgery				0.272
Conserving surgery	69 (56.1)	329 (50.7)	398 (51.6)	
Mastectomy	54 (43.9)	320 (43.9)	374 (48.4)	
ER status				0.117
Negative	69 (56.1)	314 (48.4)	383 (49.6)	
Positive	54 (43.9)	335 (51.6)	389 (50.4)	
PR status				0.143
Negative	82 (66.7)	387 (59.6)	469 (60.8)	
Positive	41 (33.3)	262 (40.4)	303 (39.2)	
HER2 status				0.219
Negative	76 (61.8)	438 (67.5)	514 (66.6)	
Positive	47 (38.2)	211 (32.5)	140 (33.5)	
Pathologic tumor stage				0.028
ypT0-is	36 (29.2)	124 (19.1)	160 (20.7)	
ypT1	43 (35.0)	205 (31.6)	248 (32.1)	
ypT2	29 (23.6)	178 (27.4)	207 (26.8)	
ypT3 ypT4 Pathologic node stage ypN0 ypN1 ypN2 ypN3	14 (11.4) 1 (0.8) 69 (56.1) 39 (31.7) 15 (12.2)	131 (20.2) 11 (1.7) 216 (33.3) 219 (33.7) 120 (18.5) 94 (14.5)	145 (18.8) 12 (1.6) 285 (39.9) 258 (33.4) 165 (17.5) 94 (12.2)	<0.001
Adjuvant Radiotherapy				0.016
Absent	14 (11.4)	36 (5.5)	50 (6.5)	
Present	109 (88.6)	613 (94.5)	722 93.5	

BMI = body mass index; ER = estrogen receptor; PR = progesterone receptor; HER2 = human epidermal growth factor-2.

**Table 2 jcm-11-06564-t002:** Characteristics of the pathologic node-negative patients’ group according to lymph node dissection number.

Variable	<10 Nodes (*n* = 69) No. (%)	≥10 Nodes (*n =* 216) No. (%)	Total (*n* = 285) No. (%)	*p*-Value
Age (years)	45.6 ± 10.1	45.4 ± 9.7	45.5 ± 9.7	0.867
BMI	23.5 ± 2.9	24.2 ± 3.4	24.0 ± 3.3	0.123
Type of surgery				0.697
Conserving surgery	41 (59.4)	134 (62.0)	175 (61.4)	
Mastectomy	28 (40.6)	82 (37.9)	110 (38.6)	
ER status				0.684
Negative	45 (65.2)	135 (62.5)	180 (63.2)	
Positive	24 (34.8)	81 (37.5)	105 (36.8)	
PR status				0.997
Negative	49 (71.0)	153 (70.8)	202 (70.1)	
Positive	20 (29.0)	63 (29.2)	83 (29.1)	
HER2 status				0.997
Negative	40 (58.0)	125 (57.9)	165 (57.9)	
Positive	29 (42.0)	91 (42.1)	120 (42.1)	
Pathologic tumor stage				0.854
ypT0-is	30 (43.5)	93 (43.0)	123 (43.2)	
ypT1	25 (36.2)	68 (31.5)	93 (32.6)	
ypT2	11 (15.9)	38 (17.6)	49 (17.2)	
ypT3	3 (4.3)	16 (7.4)	19 (6.7)	
ypT4	0 (0.0)	1 (0.5)	1 (0.3)	
Radiotherapy				0.324
Absent	10 (14.5)	22 (10.2)	32 (11.2)	
Present	59 (85.5)	194 (89.8)	253 (88.8)	

BMI = body mass index; ER = estrogen receptor; PR = progesterone receptor; HER2 = human epidermal growth factor-2.

**Table 3 jcm-11-06564-t003:** Characteristics of the pathologic node-positive patients’ group according to lymph node dissection number.

Variable	<10 Nodes (*n* = 54) No. (%)	≥10 Nodes (*n =* 433) No. (%)	Total (*n* = 487) No. (%)	*p*-Value
Age (years)	46.15 ± 10.14	44.96 ± 9.26	45.09 ± 9.36	0.379
BMI	23.83 ± 3.38	23.98 ± 3.45	23.85 ± 3.46	0.202
Type of surgery				0.343
Conserving surgery	28 (51.85)	195 (45.03)	223 (45.79)	
Mastectomy	26 (48.15)	238 (54.97)	264 (54.21)	
ER status				0.663
Negative	24 (44.44)	179 (41.34)	203 (41.68)	
Positive	30 (55.56)	254 (58.66)	284 (58.32)	
PR status				0.325
Negative	33 (61.11)	234 (54.04)	267 (54.83)	
Positive	21 (38.89)	199 (45.96)	220 (45.17)	
HER2 status				0.388
Negative	36 (66.67)	313 (72.29)	349 (71.66)	
Positive	18 (33.33)	120 (27.71)	138 (28.34)	
Pathologic tumor stage				0.737
ypT0-is	6 (11.11)	31 (7.16)	37 (7.60)	
ypT1	18 (33.33)	137 (31.64)	155 (31.83)	
ypT2	18 (33.33)	140 (32.33)	158 (32.44)	
ypT3	11 (20.37)	115 (2.56)	126 (25.87)	
ypT4	1 (1.85)	10 (2.31)	11 (2.26)	
Radiotherapy				0.125
Absent	4 (7.41)	14 (3.23)	18 (3.70)	
Present	50 (92.59)	419 (96.77)	469 (96.30)	

BMI = body mass index; ER = estrogen receptor; PR = progesterone receptor; HER2 = human epidermal growth factor-2.

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
