# Peer review of "Is Pathologic Axillary Staging Valid If Lymph Nodes Are Less than 10 with Axillary Lymph Node Dissection after Neoadjuvant Chemotherapy?"

_jcm, 2022, doi:10.3390/jcm11216564_

Round 1

Reviewer 1 Report

Authors have presented data from a retrospective cohort of patients who had neoadjuvant chemotherapy. Study has reasonable follow up period. There is disparity in the number of patients quoted in the study in the Abstract/ Materials and methods section and results.

Even though study group has good number of patients, subset of patients with <10 LNs is small ( N=123) and most of other subset analysis involves smaller number of patients in various groups. DFS and OS are dependent on a number of factors such as type of systemic therapy, compliance with adjuvant treatment, type of adjuvant treatment and rate of pCR/ residual cancer burden after neoadjuvant treatment.

1.    It is not clear from the article, whether all patients completed all intended cycles of NACT. Whether there is any early termination due to side effects/ toxicity?

2.    What proportion of patients achieved pCR in each group?

3.    No data available on the details of RT to SCF/Internal Mammary chain. Does the table shows patients who had adjuvant RT to breast?

4.    No data available on the residual nodal burden. Even if the yield is small on clearance, if most of the nodes are involved in < 10 node group, potentially this subset has worse prognosis than smaller number of nodes with residual cancer in >10 node group.

5.    No data available on patient compliance with adjuvant treatment such as endocrine treatment.

Conclusion needs to be modified to acknowledge above limitations and clearly this observation needs validation with prospective trials with broader spectrum of variables which can potentially impact DFS and OS.

Author Response

Response to Reviewer 1 Comments

Authors have presented data from a retrospective cohort of patients who had neoadjuvant chemotherapy. Study has reasonable follow up period. There is disparity in the number of patients quoted in the study in the Abstract/ Materials and methods section and results.

Even though study group has good number of patients, subset of patients with <10 LNs is small ( N=123) and most of other subset analysis involves smaller number of patients in various groups. DFS and OS are dependent on a number of factors such as type of systemic therapy, compliance with adjuvant treatment, type of adjuvant treatment and rate of pCR/ residual cancer burden after neoadjuvant treatment.

Answer) Thank you for sincere and good comments. We revised the number of patients in the Methods and limitation in the Discussion. We agree DFS and OS are dependent on several factors after neoadjuvant chemotherapy.

  1. It is not clear from the article, whether all patients completed all intended cycles of NACT. Whether there is any early termination due to side effects/ toxicity?

Answer) Thank you good comments. We included patients completed all cycles of NACT. We revised Methods in the revised manuscript.

  1. What proportion of patients achieved pCR in each group?

Answer) Thank you good comments. The <10 nodes group had 29.2% breast pCR and 56.1% axillary pCR. The ≥10 nodes group had 19.1% breast pCR and 33.3% axillary pCR. We revised result in the revised manuscript.

  1. No data available on the details of RT to SCF/Internal Mammary chain. Does the table shows patients who had adjuvant RT to breast?

Answer) Thank you good comments. The patients with supraclavicular or internal mammary lymph node metastasis (2 pts <10 nodes group vs 7 pts ≥10 nodes group) had additional period RTx. The adjuvant RT was performed in all patients with breast conserving surgery or lymph node metastasis. We revised Methods in the revised manuscript.

  1. No data available on the residual nodal burden. Even if the yield is small on clearance, if most of the nodes are involved in < 10 node group, potentially this subset has worse prognosis than smaller number of nodes with residual cancer in >10 node group.

Answer) Thank you good comments. Inclusion criteria in this study were cytology proven axillary node positive before NAC and underwent ALND after NAC. Residual nodal burden was 43.9% in the < 10 node group, and 66.7% in the ≥10 nodes group. We revised Results in the revised manuscript.

  1. No data available on patient compliance with adjuvant treatment such as endocrine treatment.

Answer) Thank you good comments. We excluded patients who did not properly perform endocrine treatment or target therapy. We revised Methods in the revised manuscript.

Conclusion needs to be modified to acknowledge above limitations and clearly this observation needs validation with prospective trials with broader spectrum of variables which can potentially impact DFS and OS.

Answer) Thank you good comments. We will need validation with prospective trials with broader spectrum of variables in the future. We revised conclusion and limitation in the revised manuscript.

Reviewer 2 Report

Dear Authors,

Congrats on your MS about a retrospective analysis of a prospective cohort of patients with EBC pts submitted to NACT followed by ALND, that interrogates the prognostic significance of the number of dissected lymph nodes as a dichotomous variable (<10 vs =>10 LN). With a relative large sample size but limited by the single-center aspect of the population, it was shown that irrespective of the dissected number of LN, EBC pts submitted to NACT followed by ALND fare similarly in the long term in terms of DFS and OS. It is suggested that ypN0 and ypN2 pts, as subgroups, fare worse in case of less than 10 LN ressected. 

Several aspects of the MS warrants improvement, starting from english grammar to overall content and even citations. Below the required improvements are summarized:

"The tumor response to chemotherapy can be PREDICTIVE TO THE BENEFIT OF TAILORED ADJUVANT THERAPIES (...)" - cite create-x, katherine, olympia trial

"The extent of ALND and the number of nodes NEEDED TO BE removed (...)"

At least citation 4 doesnt support the last statement of your first paragraph in the introduction; NSABP-B32 only tells that SNB negative patients do as well as SNB negative patients submitted to additional ALND, and not that there is improved survival with more extensive nodal ressection. You either amend this citation (and 2 and 3 if applicable) or you amend the whole paragraph.

"Thus, the decrease in the number of removed lymph nodes can UNDERESTIMATE the nodal stage"

In the last paragraph of the intro, you say that for nodal STAGING, you typically require more than 10 lymph nodes. Where? In Samsung Medical Center, correct? If so, specify it or provide evidence from the literature that this is how it is normally done, because in many centers ALND is not so extensive.

In the materials and methods, you have to provide your definition for DFS and OS.

"Among 772 patients TREATED with NAC (...)"

"The <10 nodes group had LOWER BMI, radiotherapy RECIPIENT rate, AND pathologic stage (...)"

"ACCORDING TO BREAST CANCER SUBTYPES, triple negative (...)"

"The number of dissected lymph nodes (...) which may not always be undergone with an inadequate ALND" - this sentence is extremely hard to understand. Did you mean to say that patients who had NAC may be under-sampled in the axilla?

"One study focused on the LNR (...)" what is LNR?? Not all your readers are going to be breast surgeons to know this.

Why the number of ressected nodes seemed to matter only for ypN0 and ypN2 subgroups, such different subgroups from a general prognostic perspective? Any biologically-supported hypothesis for this? Or is this just a chances finding from a subgroup analysis? You have to make a comment on this, and for me this looks like just chances playing a trick...

Your conclusion should be totally rewired. Your main finding here, in the overall study population, is that EBC patients submitted to NAC followed by ALND where less than 10 nodes where ressected do as well as those with more than 10 nodes. This is important in the sense that it is reassuring for patients and surgeons that less than 10 nodes can be removed, with equivalent clinical outcomes and potentially less impact on the QoL and AE profile typically seen with ALND (i.e. lymphedema), that were not analyzed in this study (yet another weakness of your study that you have to make explicit). Your subgroup findings are not supportive of your conclusions, and are probably due to chances, as they dont find any biological rationale (if it was only ypN0 pts doing fine and ypN1 and ypN2 pts doing worse in case of less than 10 nodes removed, your subgroup conclusions would have made sense...) 

Author Response

(The authors gave the same response as above.)

Reviewer 3 Report

Title

Should be rephrased as a question ie Is pathologic axillary staging valid if …….? If it is a question.

Abstract

Line 23 prognostic vaue of the number of lymph nodes … removed …. In breast cancer patients etc

Line 26 ‘cytology proven axillary node positive cytology proven axillary node’ – sentence contains duplicate. Suggest ‘ cytology proven involved axillary nodes at diagnosis.’

Line 30  adjust English syntax of sentence.

Line 33 I would also quote the hRs and p values for N1 and N3 so the abstract reader gets a full sense of the data.

Line 36-37 ‘might need more medical attention’ – this is a very vague phrase, try and be more specific regarding action eg ‘might be considered for additional staging/ closer surveillance.

Could consider reporting whether the location of relapse between <10 LN and >=10 LN.

The three hypotheses for a link are either; 1) that lower LN population allows metastases to reach the systemic circulation earlier, 2) that involved LNs may have been left behind by less complete surgery, and 3) that missing positive nodes leads to less post-op adjuvant therapy.

Introduction

42 ‘The tumor response to chemotherapy can be predicted’. A slightly odd phrase. Obviously the local tumour response can be measured directly, not predicted. I presume you mean ‘The long term effectiveness of chemotherapy can be predicted’?

49 ‘Previous studies have suggested improved survival with more extensive nodal dissections in both node-negative and node-positive patients [2-4]’. Ref 2 relates to pCR and prognosis in NAC, I cannot find a mention of LN harvest and outcome. Similarly, although ref 4 is about ALND I can find no reference to LN yield and prognosis. I haven’t checked ref 3 – check refs please

The most relevant reference appears to be - DOI: 10.1007/s10549-019-05500-9, ‘Axillary lymph node dissection in node-positive breast cancer: are ten nodes adequate and when is enough, enough?’
This study appears to be modelled on the above ref but the above ref is not cited – suggest it should be.

53 word missing - ‘after NAC …is…7.8%’.

53 ‘in overall survival (OS, p=0.2) and the’ should read ‘in overall survival (OS, p=0.2) or the’.

60 ‘Thus, the decrease in the number of removed lymph nodes can be underestimated in the nodal stage’ – sentence does not make sense.

Methods

Methods mention 418 patients, results mention 772 – which is correct?

Suggest consort diagram to show excluded patients

Results

Did patients receive post-op chemo – either capecitabine, T-DM1 or other? If so I would suggest including in table 1.

What were the pCR rates in the two groups? Presumptively much greater in the <10 LN group

92 ‘lower BMI’ rather than ‘less BMI’.

95 – table 1 –I would suggest a multivariate of the whole patient group, at least compensating for T and N stage would be appropriate as the >10 LN patients are clearly a higher risk cohort by this parameter to accompany fig 1

Why no table for the LN positive patients like table Table, they could be incorporated in the same table possibly, as you have KM curves for them.

Noting from fig 2a that there were more relapses but not more deaths – could the sites of relapse be given ie local, ax LN or distant

112 ‘There was no significant difference in DFS (p=0.024) and OS (p=0.024) between the <10 nodes group and the >= 10 lymph nodes group ….’. These are not the p values on the graphs in Fig 2B which are p=0.846 (DfS) and 0.774 (OS). I presume this is a typo – obviously p=0.024 is significant.

ypN1 and ypN2 for <10 LNs are low numbers (n=39 and 15), could analyze together.

Discussion

24 - ref 9 is about the predictive power of pCR, not ALND adequacy – check refs please

there is a notable increase in ypN2 and n3 in >10 LN patients., they are higher stage patients but this is not mentioned. They should try and explain why there is such a difference given the biology of the tumours is the same.

Should contrast to the Rosenberger paper I list earlier where LN yield as a whole correlated with outcome. However, numbers were much larger.

136 – ‘which may not always be undergone with an inadequate ALND.’ Needs rephrasing, also do you mean ‘adequate ALND’ rather than ‘inadequate’?

154 – ‘The > 10 nodes group might contain small positive lymph node.’ What do you mean here. Do you mean ‘might harbour occult, unresected positive lymph nodes’.

161 ‘was introduced in 2017 San Antonio breast cancer symposium’. Do you mean ‘was presented at the 2017 San Antonio breast cancer symposium’? This paragraph is very unclear. Is this trial underway? It needs explaining more clearly.

166 – ‘The <10 group might contain some N3 group actually.’ It is unclear what this sentence means.

You do not make any attempt to explain your results that I can find. You have noted that when compared in LN burden groups  patients with <10 LN do worse – why do you think this is? Also, why do you think patients with > 10 LN taken have higher LN burdens – I presume it is because positive nodes are easier to find at surgery but it then should have the opposite effect on outcome.

Three hypotheses for a link to worse outcome could be; 1) that lower LN population allows metastases to reach the systemic circulation earlier, 2) that involved LNs may have been left behind by less complete surgery, and 3) that missing positive nodes leads to less post-op adjuvant therapy.

Should mention that there is more chance after NAC that distribution of LN involvement may be more variable due to variable response in different nodes.

175 – ‘might need more medical attention’. What actually should we do differently for these patients. Authors should suggest some specific actions, this is very vague.

Author Response

Response to Reviewer 3 Comments

Title

Should be rephrased as a question ie Is pathologic axillary staging valid if …….? If it is a question.

 Answer) Thank you for sincere and good comments. We revised title as a question.

Abstract

Line 23 prognostic vaue of the number of lymph nodes … removed …. In breast cancer patients etc

Answer) Thank you good comments. We included “removed” in this sentence. We revised abstract in the revised manuscript

Line 26 ‘cytology proven axillary node positive cytology proven axillary node’ – sentence contains duplicate. Suggest ‘ cytology proven involved axillary nodes at diagnosis.’

Answer) Thank you good comments. As your suggestion, we revised this sentence in the revised manuscript

Line 30  adjust English syntax of sentence.

Answer) Thank you good comments. We revised syntax of this sentence in the revised manuscript

Line 33 I would also quote the hRs and p values for N1 and N3 so the abstract reader gets a full sense of the data.

Answer) Thank you good comments. We added p values for ypN1. The <10 nodes group cannot have ypN3. We revised abstract in the revised manuscript

Line 36-37 ‘might need more medical attention’ – this is a very vague phrase, try and be more specific regarding action eg ‘might be considered for additional staging/ closer surveillance.

Answer) Thank you good comments. As your suggestion, we revised this phrase in the revised manuscript

Could consider reporting whether the location of relapse between <10 LN and >=10 LN.

Answer) Thank you good comments. In the <10 lymph nodes group, there were 31 recurrences, of which 16 had local recurrence, in the ≥10 lymph nodes group , 153 had recurrence, and 68 had local recurrence. We add Results in the revised manuscript.

The three hypotheses for a link are either; 1) that lower LN population allows metastases to reach the systemic circulation earlier, 2) that involved LNs may have been left behind by less complete surgery, and 3) that missing positive nodes leads to less post-op adjuvant therapy.

Answer) Thank you good comments. We added this hypotheses in Discussion.

Introduction

42 ‘The tumor response to chemotherapy can be predicted’. A slightly odd phrase. Obviously the local tumour response can be measured directly, not predicted. I presume you mean ‘The long term effectiveness of chemotherapy can be predicted’?

Answer) Thank you good comments. We mean ‘The long term effectiveness of chemotherapy can be predicted’ We revised this sentence in revised manuscript..

49 ‘Previous studies have suggested improved survival with more extensive nodal dissections in both node-negative and node-positive patients [2-4]’. Ref 2 relates to pCR and prognosis in NAC, I cannot find a mention of LN harvest and outcome. Similarly, although ref 4 is about ALND I can find no reference to LN yield and prognosis. I haven’t checked ref 3 – check refs please

The most relevant reference appears to be - DOI: 10.1007/s10549-019-05500-9, ‘Axillary lymph node dissection in node-positive breast cancer: are ten nodes adequate and when is enough, enough?’
This study appears to be modelled on the above ref but the above ref is not cited – suggest it should be.

Answer) Thank you good comments. As your suggestion, we revised reference and Introduction in revised manuscript.

53 word missing - ‘after NAC …is…7.8%’.

Answer) Thank you good comments. We added the word in the Introduction.

53 ‘in overall survival (OS, p=0.2) and the’ should read ‘in overall survival (OS, p=0.2) or the’.

Answer) Thank you good comments. We revised the sentence in the Introduction.

60 ‘Thus, the decrease in the number of removed lymph nodes can be underestimated in the nodal stage’ – sentence does not make sense.

Answer) Thank you good comments. We revised the sentence in the Introduction.

Methods

Methods mention 418 patients, results mention 772 – which is correct?

Answer) Thank you good comments. 772 patients is correct. We revised the methods in the revised manuscript.

Suggest consort diagram to show excluded patients

Answer) Thank you good comments. We made consort diagram as Figure 1 in the revised manuscript.

Results

Did patients receive post-op chemo – either capecitabine, T-DM1 or other? If so I would suggest including in table 1.

Answer) Thank you good comments. Until 2015, in our hospital, these treatment (capecitabine, T-DM1) was not implemented

What were the pCR rates in the two groups? Presumptively much greater in the <10 LN group

Answer) Thank you good comments. The <10 nodes group had 29.2% breast pCR and 56.1% axillary pCR. The ≥10 nodes group had 19.1% breast pCR and 33.3% axillary pCR. We revised result in the revised manuscript.

92 ‘lower BMI’ rather than ‘less BMI’.

Answer) Thank you good comments. We revised the sentence in the Results.

95 – table 1 –I would suggest a multivariate of the whole patient group, at least compensating for T and N stage would be appropriate as the >10 LN patients are clearly a higher risk cohort by this parameter to accompany fig 1

Answer) Thank you good comments. We present our initial analysis in figure 2-A. The compensating analysis of T and N stages in figure 2-B. Compensating results showed DFS (p=0.056) and OS (p=0.528). In compensating result, there are slight trend in DFS, however there are no significant difference. We added Figure 2-A,B in the revised manuscript.

Why no table for the LN positive patients like table Table, they could be incorporated in the same table possibly, as you have KM curves for them.

Answer) Thank you good comments. We analyzed 487 patients with LN positive. There were no significant difference DFS (p=0.972) and OS (p=0.758) in LN positive patients. We added Table 3 and Figure 3 in the Results.

Noting from fig 2a that there were more relapses but not more deaths – could the sites of relapse be given ie local, ax LN or distant

Answer) Thank you good comments. In ypN0 subgroup, there were 13 recurrences in the <10 lymph nodes group, of which 5 had local recurrence, 26 had recurrence in the ≥10 lymph nodes group, and 7 had local recurrence. We add Results in the revised manuscript.

112 ‘There was no significant difference in DFS (p=0.024) and OS (p=0.024) between the <10 nodes group and the >= 10 lymph nodes group ….’. These are not the p values on the graphs in Fig 2B which are p=0.846 (DfS) and 0.774 (OS). I presume this is a typo – obviously p=0.024 is significant.

Answer) Thank you good comments. This is a typo. We revised Results in the revised manuscript.

ypN1 and ypN2 for <10 LNs are low numbers (n=39 and 15), could analyze together.

 Answer) Thank you good comments. We analyzed patients with LN positive (ypN1, ypN2). There were no significant difference DFS (p=0.972) and OS (p=0.758) in LN positive patients. We added Figure 3 in the Results.

Discussion

24 - ref 9 is about the predictive power of pCR, not ALND adequacy – check refs please

 Answer) Thank you good comments. We revised reference in the revised manuscript.

there is a notable increase in ypN2 and n3 in >10 LN patients., they are higher stage patients but this is not mentioned. They should try and explain why there is such a difference given the biology of the tumours is the same.

Answer) Thank you good comments. There is a notable increase in ypN2 and ypN3 in the ≥10 LN patients. In the group with the <10 lymph nodes, there is no ypN3 and it is possible that axillary staging was underestimated overall. We revised discussion in the revised manuscript.

Should contrast to the Rosenberger paper I list earlier where LN yield as a whole correlated with outcome. However, numbers were much larger.

Answer) Thank you good comments. Rosenberger study were estimated to be 1–7, 8–22, and > 22 LNs after NAC. The better survival was independently associated with retrieval of up to approximately 20 LN[3]. In our study, 10 lymph nodes were divided based on the criteria, and in the case of ypN2, there was also a difference in the survival rate. This study showed that worse survival was associated with retrieving fewer LNs, likely as a result of an under-staged axilla and missed opportunity for adjuvant therapy. We revised discussion in the revised manuscript.

136 – ‘which may not always be undergone with an inadequate ALND.’ Needs rephrasing, also do you mean ‘adequate ALND’ rather than ‘inadequate’?

Answer) Thank you good comments. We revised sentence in the revised manuscript.

154 – ‘The > 10 nodes group might contain small positive lymph node.’ What do you mean here. Do you mean ‘might harbour occult, unresected positive lymph nodes’.

Answer) Thank you good comments. We mean ‘might harbour occult, unresected positive lymph nodes’. We revised sentence in the revised manuscript.

161 ‘was introduced in 2017 San Antonio breast cancer symposium’. Do you mean ‘was presented at the 2017 San Antonio breast cancer symposium’? This paragraph is very unclear. Is this trial underway? It needs explaining more clearly.

Answer) Thank you good comments. We mean ‘was presented at the 2017 San Antonio breast cancer symposium’. We revised sentence in the revised manuscript.

166 – ‘The <10 group might contain some N3 group actually.’ It is unclear what this sentence means.

 Answer) Thank you good comments. The <10 nodes group might be higher axillary stage actually. We revised sentence in the revised manuscript.

You do not make any attempt to explain your results that I can find. You have noted that when compared in LN burden groups  patients with <10 LN do worse – why do you think this is? Also, why do you think patients with > 10 LN taken have higher LN burdens – I presume it is because positive nodes are easier to find at surgery but it then should have the opposite effect on outcome.

Answer) Thank you good comments. We think that worse survival in the <10 nodes group was associated with retrieving fewer LNs, likely as a result of an under-staged axilla and missed opportunity for adjuvant therapy. Also we agree that patients with the ≥ 10 LN taken have higher LN burdens because positive nodes are easier to find at surgery but it then should have the opposite effect on outcome. We revised sentence in the revised manuscript.

Three hypotheses for a link to worse outcome could be; 1) that lower LN population allows metastases to reach the systemic circulation earlier, 2) that involved LNs may have been left behind by less complete surgery, and 3) that missing positive nodes leads to less post-op adjuvant therapy.

Should mention that there is more chance after NAC that distribution of LN involvement may be more variable due to variable response in different nodes.

Answer) Thank you good comments. We added this hypotheses and discussion in revised manuscript.

175 – ‘might need more medical attention’. What actually should we do differently for these patients. Authors should suggest some specific actions, this is very vague.

Answer) Thank you good comments. breast cancer patients with less than 10 lymph nodes number in ALND after NAC might be considered for additional staging or closer surveillance when compared to patients with 10 or more than lymph node when compared to patients with 10 or more than lymph node in ypN0 and ypN2 subgroups. We revised conclusion in revised manuscript.

Round 2

Reviewer 1 Report

Many thanks for your comments and revisions.

1.    RT is an effective treatment for axillary nodal metastasis and in many centres this is done as an alternative to clearance in a good proportion of suitable patients. Hence it is important to get clarity on this treatment.

Any data available on patients who had RT to SCF and IMC. Authors suggest that patients had additional RT when IMC/SCF nodes positive. However, in most centres if Axillary nodal clearance shows 4 or more nodes with metastasis, RT is given to SCF and possibly to IMC chain after ANC. What was the indication for SCF and IMC irradiation in the current study? And what proportion of patients had RT to SCF and IMC?

2.    Authors have given residual nodal burden as 43.9% and 66.7% in the <10 and >10 node groups. However this is the proportion of patients with residual nodal disease after NACT. What would be more useful is the data on the proportion of nodes positive out of total nodal harvest in each group ( residual nodal burden). Eg: a patient with 7/9 nodes positive ( belong to <10 node group) may have poor outcome in comparison to 2/ 18 nodes positive patient ( > 10 node group)

3.    With the additional data on recurrence, it is not clear whether the local recurrence was in breast recurrence or recurrence in the axilla?

Author Response

Response to Reviewer 1 Comments

Many thanks for your comments and revisions. 

  1. RT is an effective treatment for axillary nodal metastasis and in many centres this is done as an alternative to clearance in a good proportion of suitable patients. Hence it is important to get clarity on this treatment.

Any data available on patients who had RT to SCF and IMC. Authors suggest that patients had additional RT when IMC/SCF nodes positive. However, in most centres if Axillary nodal clearance shows 4 or more nodes with metastasis, RT is given to SCF and possibly to IMC chain after ANC. What was the indication for SCF and IMC irradiation in the current study? And what proportion of patients had RT to SCF and IMC?

Answer) Thank you for sincere and good comments. In our study, the indications of RT to SCF were T3 or higher, N2 or higher, or SCN metastasis confirmed by FNA. The indications of RT to IMC are metastasis suspected by breast MRI or metastasis confirmed by FNA. (RT to SCF: 21 pts (17.1%) <10 nodes group vs 225 pts (34.6%) ≥10 nodes group, RT to IMC: 6 pts (4.9%) <10 nodes group vs 27 pts (4.2%) ≥10 nodes group). We revised Material and Methods in the revised manuscript.

  1. Authors have given residual nodal burden as 43.9% and 66.7% in the <10 and >10 node groups. However this is the proportion of patients with residual nodal disease after NACT. What would be more useful is the data on the proportion of nodes positive out of total nodal harvest in each group ( residual nodal burden). Eg: a patient with 7/9 nodes positive ( belong to <10 node group) may have poor outcome in comparison to 2/ 18 nodes positive patient ( > 10 node group)

Answer) Thank you for good comments. The proportion of nodes positive out of total nodal harvest is 1.4/8.6 nodes positive (16.3%) in the <10 nodes group and 2.2/ 14.7 in the ≥10 nodes group (15.0%). We revised Results in the revised manuscript.

  1. With the additional data on recurrence, it is not clear whether the local recurrence was in breast recurrence or recurrence in the axilla?

Answer) Thank you for good comments. In the <10 nodes group, there were 31 recurrences, of which 16 (breast recurrence: 4, axilla recurrence: 12) had local recurrence. In the ≥10 nodes group, 153 had recurrence, of which 68 (breast recurrence: 14, axilla recurrence: 53, both recurrence: 1) had local recurrence. We revised Results in the revised manuscript.

Reviewer 3 Report

This manuscript is much improved with useful additional data and reasonable discussion of results obtained.

Some further more minor comments/corrections:

Line 49 – should be ‘nodes needing to be’ not ‘nodes needed to be’.

Line 64 – should be ‘can underestimate the nodal’ not ‘can underestimate in the nodal’

Thanks for Consort diagram. Suggest put numbers of ineligble people with the four categories on for more detail.

Line 110 – should be ‘After compensating for ypT and ypN staging’ rather than ‘In compensating ypT and ypN staste’. Also I would say that p=0.056 is a strong trend, not slight.

Line 113 no significant difference ….. in ….DFS

It is interesting that your axillary pCR (in a group of patients with proven axillary disease pre-NACT) was higher than your breast pCR. Worth a mention in your discussion if not already made.

Thank you for adding table 3.

Thanks for adding in the relapse location data

Line 159 – ‘However, in ypN0 subgroup…’. This sentence could be condensed to ‘. However, in both the ypN0 and ypN2 subgroups, the <10 nodes group had worse DFS than the >=10 nodes group.’

163 – I would expand the beginning of this sentence to ‘The hypotheses to explain the inferior outcomes observed in this study where less axillary LNs were taken include; ……’

192 – ‘focusing’ not ‘focused’

200 – ‘can have ALND omitted’          

203 ‘shows’ not ‘show’

204 suggest ‘ the <10 nodes group may have higher axillary stage reality than apparent from the more limited surgery.’

205 – suggest ‘Another study compared groups where 1–7, 8–22, and > 22 LNs were harvested after NAC.’

208 – ‘based on the criteria’ – which criteria are you referring to here.

217 ‘ is dependant’

222 suggest ‘insight into the significance of the total lymph node number obtained at ALND following NAC.’

Author Response

Response to Reviewer 3 Comments

This manuscript is much improved with useful additional data and reasonable discussion of results obtained.

Some further more minor comments/corrections:

Line 49 – should be ‘nodes needing to be’ not ‘nodes needed to be’.

Answer) Thank you for good comments. We revised the sentence in the revised manuscript.

Line 64 – should be ‘can underestimate the nodal’ not ‘can underestimate in the nodal’

Answer) Thank you for good comments. We revised the sentence in the revised manuscript.

Thanks for Consort diagram. Suggest put numbers of ineligble people with the four categories on for more detail.

Answer) Thank you for good comments. We revised Figure 1 in the revised manuscript.

Line 110 – should be ‘After compensating for ypT and ypN staging’ rather than ‘In compensating ypT and ypN staste’. Also I would say that p=0.056 is a strong trend, not slight.

Answer) Thank you for good comments. We agree your opinion. We revised the sentence in the revised manuscript.

Line 113 no significant difference ….. in ….DFS

Answer) Thank you for good comments. We revised the sentence in the revised manuscript.

It is interesting that your axillary pCR (in a group of patients with proven axillary disease pre-NACT) was higher than your breast pCR. Worth a mention in your discussion if not already made.

Answer) Thank you for good comments. We revised Discussion in the revised manuscript.

Thank you for adding table 3.

Thanks for adding in the relapse location data

Line 159 – ‘However, in ypN0 subgroup…’. This sentence could be condensed to ‘. However, in both the ypN0 and ypN2 subgroups, the <10 nodes group had worse DFS than the >=10 nodes group.’

Answer) Thank you for good comments. We revised the sentence in the revised manuscript.

163 – I would expand the beginning of this sentence to ‘The hypotheses to explain the inferior outcomes observed in this study where less axillary LNs were taken include; ……’

Answer) Thank you for good comments. We agree your opinion. We revised the sentence in the revised manuscript.

192 – ‘focusing’ not ‘focused’

Answer) Thank you for good comments. We revised the sentence in the revised manuscript.

200 – ‘can have ALND omitted’         

Answer) Thank you for good comments. We revised the sentence in the revised manuscript. 

203 ‘shows’ not ‘show’

Answer) Thank you for good comments. We revised the sentence in the revised manuscript. 

204 suggest ‘ the <10 nodes group may have higher axillary stage reality than apparent from the more limited surgery.’

Answer) Thank you for good comments. We revised the sentence in the revised manuscript. 

205 – suggest ‘Another study compared groups where 1–7, 8–22, and > 22 LNs were harvested after NAC.’

Answer) Thank you for good comments. We revised the sentence in the revised manuscript. 

208 – ‘based on the criteria’ – which criteria are you referring to here.

Answer) Thank you for good comments. We divided LN into 10, which is the standard of N3 of the NCCN standard. We revised Discussion in the revised manuscript.

217 ‘ is dependant’

Answer) Thank you for good comments. We revised the sentence in the revised manuscript. 

222 suggest ‘insight into the significance of the total lymph node number obtained at ALND following NAC.’

Answer) Thank you for good comments. We revised the sentence in revised manuscript.
